# The effect of music on body sway when standing in a moving virtual environment

**Shaquitta Dent**[1], **Kelley Burger**[1], **Skyler Stevens**[1], **Benjamin D. Smith**[2‡], **Jefferson W. Streepey**[1‡*]

**1** Department of Kinesiology, School of Health and Human Sciences, Indiana University-Purdue University Indianapolis, Indianapolis, Indiana, United States of America, **2** Music and Arts Technology, Computer Information and Graphics Technology, Purdue School of Engineering & Technology, Indiana University-Purdue University Indianapolis, Indianapolis, Indiana, United States of America

☯ These authors contributed equally to this work.
‡ These authors also contributed equally to this work.
* jwstreep@iupui.edu

**Data Availability Statement:** https://doi.org/10.13026/x32c-cz47.

**Funding:** The authors received no specific funding for this work.

## Abstract

Movement of the visual environment presented through virtual reality (VR) has been shown to invoke postural adjustments measured by increased body sway. The effect of auditory information on body sway seems to be dependent on context with sounds such as white noise, tones, and music being used to amplify or suppress sway. This study aims to show that music manipulated to match VR motion further increases body sway. Twenty-eight subjects stood on a force plate and experienced combinations of 3 visual conditions (VR translation in the AP direction at 0.1 Hz, no translation, and eyes closed) and 4 music conditions (Mozart's Jupiter Symphony modified to scale volume at 0.1 Hz and 0.25 Hz, unmodified music, and no music) Body sway was assessed by measuring center of pressure (COP) velocities and RMS. Cross-coherence between the body sway and the 0.1 Hz and 0.25 Hz stimuli was also determined. VR translations at 0.1 Hz matched with 0.1Hz shifts in music volume did not lead to more body sway than observed in the no music and unmodified music conditions. Researchers and clinicians may consider manipulating sound to enhance VR induced body sway, but findings from this study would not suggest using volume to do so.

## Introduction

The use of virtual reality (VR) for the study of body sway has been in practice for well over a decade. VR provides an immersive, controlled visual environment that can be used to expose subjects to visual motion as virtual walls or objects pass back and forth in the periphery to increase body sway during stance. In these studies the nature of body sway takes on the characteristics of the visual motion in terms of both amplitude and frequency [1–3]. Furthermore, these effects on body sway can be influenced when visual motion presented through VR is combined with the manipulation of other sensory information used to regulate standing balance [4–6].

**Competing interests:** The authors have declared that no competing interests exist.

While the contributions of visual, vestibular, and somatosensory inputs on body sway have been well studied, the effects of different auditory inputs have not been conclusively demonstrated. It has been shown that sway path and sway variability appear to decrease when white noise is supplied to standing subjects [7]. However other studies have shown that constant tones and background conversation lead to increased body sway [8, 9]. When the provided auditory cue was music as opposed to white noise or tones, Palm et al. [10] showed that body sway was not differently affected. Others have suggested that the type of music might matter with one study demonstrating that listening to Mozart's Jupiter reduced body sway compared to other pieces of music [11] and another suggesting that increasing the groove supplied by the music would increase body sway [12].

For this study we used sound to enhance the perception of visual motion as a VR environment translated in the subject's sagittal plane so that the environment appeared to move forward and backward about the subject. To do this, we manipulated the sound of Mozart's Jupiter so that it increased and decreased in volume at the same frequency as the VR translation. We hypothesized that adding such an audio cue would increase the changes in body sway observed with VR visual motion.

## Methods

### Subject recruitment

Twenty-eight healthy college students (13 men; 15 women), between 18 and 35 years of age participated. Subjects were excluded from the experiment if they had a history of vertigo, motion sickness, or vestibular deficits. The experimental protocol was approved by the Indiana University Institutional Review Board and was performed in accordance with the Declaration of Helsinki. All subjects gave their informed consent before inclusion in the study.

### Experimental set-up

Subjects wore an Oculus Rift headset (Oculus VR, LLC., Irvine, CA) which presented a stereoscopic, three-dimensional, immersive virtual reality environment. The simulation was controlled by an interactive application authoring software, Unity 5.0 (Unity Technologies, San Francisco, CA) that responded in real-time to rotational and positional motion of the Oculus Rift to naturally update the visual environment with head motion. As a consequence, while wearing the headset, subjects were able to move their heads from left to right, upward and downward, and objects within the virtual reality environment retained their true perspective and position in space. The virtual reality environment simulated by the headset was a two-way street with four story buildings and store fronts lining the left and right side of the street. A clear blue sky with scattered clouds could also be viewed above the horizon (Fig 1). On-ear headphones integrated into the Oculus Rift headset were used to provide a selection of Mozart's Jupiter symphony to the subjects. Jupiter was chosen as it has previously been shown to alter body sway during stance [11]. Music playback was controlled by Max 7.3.5 application (Cycling '74, Walnut, CA), an audio editing application that can compile with Unity to distort sounds in virtual simulations.

Subjects were asked to wear the Oculus Rift headset and stand barefoot atop a balance plate (Bertec Corp., Columbus, OH) with their heads looking forward, feet shoulder width apart, and hands by their side. Once the headset was placed on the subject it was not removed until the experiment was over. During this time the subjects only viewed the virtual environment including when they were between trials. When standing quietly, subjects stood either with their eyes closed or open looking at the VR environment. A third visual condition also existed where visual motion of the virtual reality environment produced through natural head motion

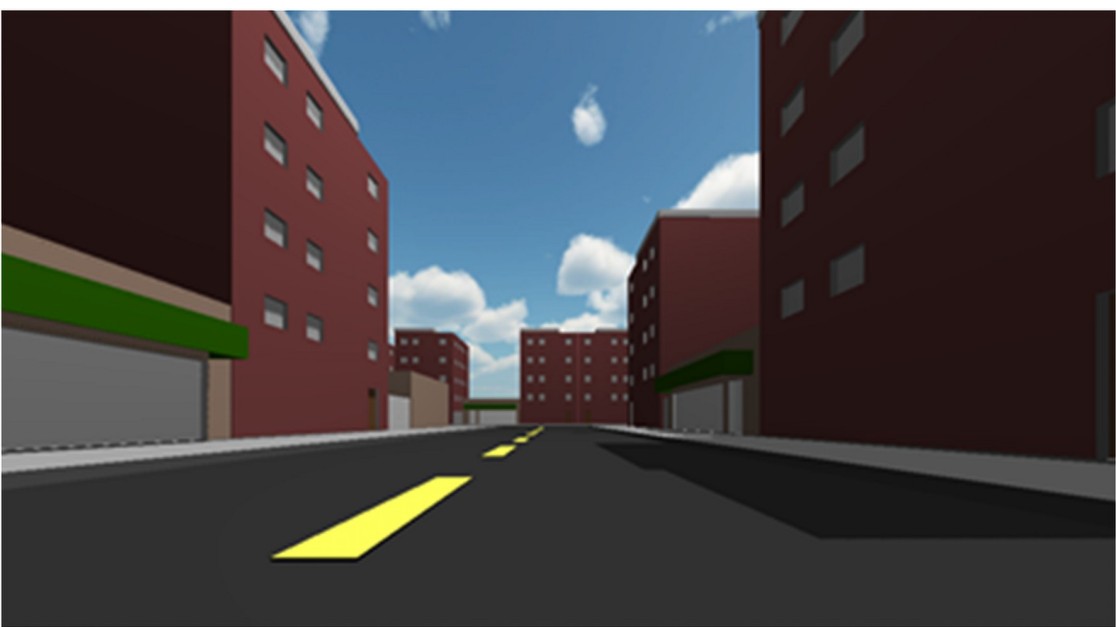

**Fig 1. A view of the virtual reality environment displayed in the Oculus Rift.** For the 0.1 Hz scene translation condition, the street and buildings would translate forward and backward to give the viewer the illusion of motion.

was combined with sinusoidal translation (± 5 m) of the environment in the anterior-posterior (AP) direction at a frequency of 0.1 Hz. AP translation of the scene was chosen because it has been well documented that such visual motion can invoke a postural response [13, 14]. Three music conditions were also introduced to the subjects, a condition where the volume of the music shifted from low to high at 0.1 Hz, a condition where it shifted at 0.25 Hz, and a condition where it played at constant volume. Volume was self-selected by the subjects so that it was never excessively loud but could always be heard. There was also a fourth audio condition where no music was played and no other sounds were provided. In total, subjects completed 12 trials of standing while experiencing combinations of visual and music stimulations. In the conditions where there were both scene translation and music volume shifting, the signals started simultaneously so that the reversal points of the 0.1 Hz music volume shifting matched the reversal points of the scene translation and that the volume scaled up as the subjects viewed forward motion and the volume scaled down as backwards motion was viewed. Trials were presented to the subjects in random order and lasted 60s each.

### Data collection and analysis

Body sway over the course of the 60s trials was measured from center of pressure (COP) data collected from the balance plate sampling at 1000 Hz. A custom Matlab (Mathworks, Natick, MA) program was used to low-pass filter the COP data at 4 Hz using a 4th order butterworth filter. The program analyzed the COP data in the AP and medial-lateral (ML) directions. AP and ML COP average velocities were determined. Root mean square (RMS) of the AP and ML COP data was calculated for six 10s intervals of each trial and then averaged across the six intervals. The area of body sway was determined using the method and adapted code first described by Duarte and Zatsiorsky [15]. Finally, a cross-spectral coherence analysis was performed to determine the amount of correlation between the ML and AP and the 0.1 Hz VR and music manipulations as well as the 0.25 Hz music input. Coherence estimates close to 1.0

suggest a near perfect synchronization between two time series and estimates closer to 0.0 suggest no relationship between the time series data [16, 17].

A 3x4 (visual stimulus x music) ANOVA with repeated measurements was performed on the COP data. A significance level of p<0.05 was used. When appropriate, Bonferroni post hoc comparisons were conducted to determine differences among the factors. All statistical analyses were done using SAS statistical software version 9.4 (SAS Institute, 2013).

## Results

Translation of the VR scene also led to increased levels of sway compared to the eyes closed and no scene translation conditions AP COP RMS [F(2,54) = 42.55, p <.0001]], AP COP velocity [F(2,54) = 89.26, p <.0001]], and average ML COP velocity [F(2,54) = 3.91, p = 0.0260)]. Further AP COP coherence at 0.1 Hz, the frequency of the scene translation was significantly elevated [F(2,54) = 59.35, p <.0001]] when the moving VR scene was viewed (Fig 2).

Listening to music with the volume shifting at 0.1 Hz increased body sway compared to the other auditory conditions (Fig 3). Significant main effects for music showed increased AP COP RMS [F(3,81) = 5.08, p = 0.0028]], and average velocity [F(3,81) = 5.64, p = 0.0015]]. Average COP velocity in the ML direction was also significantly increased in the 0.1 Hz music condition [F(3,81) = 2.79, p = 0.0455)].

With the exception of the effects of music and visual stimulus on average ML COP velocity, no other significant main effects or interactions were observed for any of the ML COP measures or on the area of the body sway.

## Discussion

Our hypothesis that adding a shifting music volume at the same frequency as the VR environment motion would increase body sway was not supported in our data.

As expected, the translating VR scene led to increased body sway in the AP direction that was strongly correlated with the visual motion stimulus. This finding is in line with previous studies examining the effect of AP optic flow on body sway [13, 14] and is consistent with the dorsiflexor and plantarflexor motion of the ankle observed during stance with the feet placed in a neutral position [18]. Matching a 0.1 Hz auditory stimulus with a translating VR scene did not appear to enhance the effect of the visual stimulation as sway was no different in these sound conditions compared to the unmodified music and music shifting volume at 0.25 Hz.

However, the inclusion of music shifting at 0.1 Hz appeared to have increased AP body sway overall as evidenced by increased RMS and average velocity regardless of the visual condition (0.1 Hz scene translation, no scene translation, eyes closed). These findings are consistent with previous research showing that a constant tone alternating at a frequency of 0.1 Hz between two speakers, a shifting in the location of the sound source, could increase body sway [9] Further there exists the possibility that this increase was facilitated by conditions where there was concurrent viewing of the moving VR scene which was correlated with body sway. More surprising was that listening to music shifting at 0.25 Hz produced body sway at the same levels observed with the unmodified music and no music conditions. Other studies have noted that auditory stimuli presented at 0.2 Hz [19] and 0.25 Hz [20] could influence body motion, but 0.25 Hz music shifting seemed to have no effect on body sway in the present study. A possible explanation for the difference in these findings is that though these studies examined how the frequency of an auditory cue affected sway, the delivery of the stimuli was quite a bit different. In one case, the auditory stimulus was a series of tones delivered via a metronome [20]. In the other the location of the sound shifted about the space occupied by the

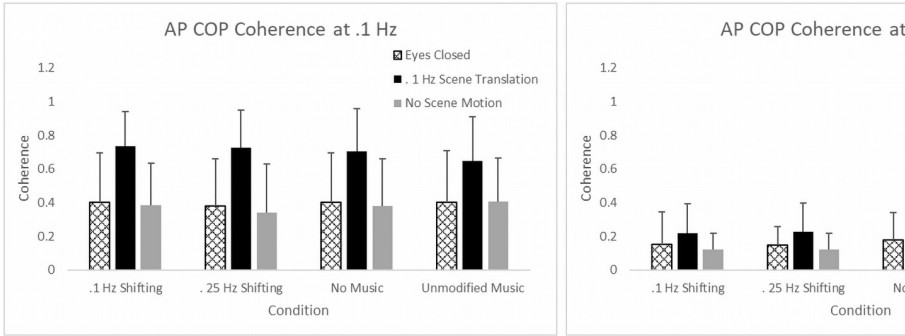

**Fig 2. AP COP Coherence at 0.1 and 0.25 Hz.** Subjects were exposed to combinations of visual conditions (eyes closed, viewing a static VR environment, viewing a VR environment with 0.1 Hz translation) and music conditions (no music, unmodified music, and music with volume shifting at 0.25 Hz and 0.1 Hz). AP COP coherence was highest when subjects experienced 0.1 Hz VR motion Music volume shifting at 0.1 Hz and 0.25 Hz appeared to have no more effect on coherence than the unmodified music and no music conditions. Error bars indicate the standard deviation.

subjects [19] in a manner that would be consistent with movement within the environment. In the present study, the sound was modified only by shifting its loudness; nothing was done to alter its location. Having the location of the music source shift along with the scene might have resulted in the expected combined effect on body sway.

It should also be noted that while the loudness of the auditory cue shifted at 0.1 Hz and 0.25 Hz, nothing was done to alter examine the effects of the rhythm chosen music on sway. As there is evidence to suggest that rhythmic cues could influence both gait [21] and sway [12], another way to examine the combined effects of the visual motion and the auditory cue of the present study would have been to match motion of the scene to the rhythm of the music. Doing so would have reduced the complexity of the task for the subjects (to stand while experiencing visual, loudness, and rhythm cues) while making use of music rhythm effect on motion.

## Conclusion

Findings from the present study do not demonstrate an increase in body sway when music loudness is modified to reinforce visual motion experienced by subjects. Though we discovered that audio cues presented at 0.1 Hz did increase sway overall compared to other audio conditions, we believe that this increase is attributable to the inclusion of 0.1Hz VR translation

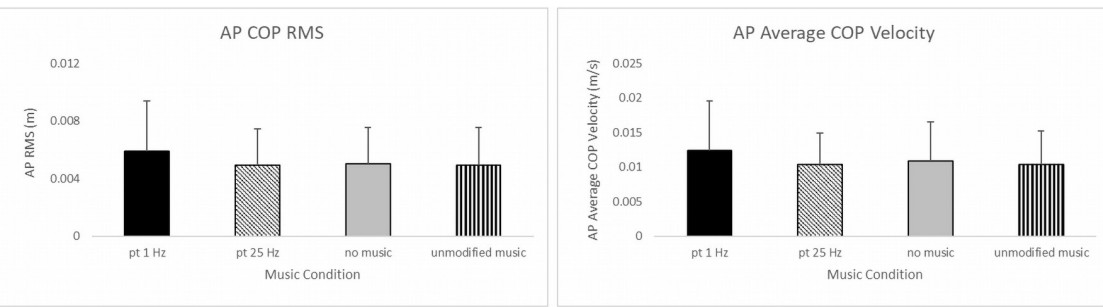

**Fig 3. The effect of music on mean AP COP RMS (left), mean and mean AP Average COP velocity (right).** Subject were exposed to no music, unmodified music, and music with volume shifting at 0.25 Hz and 0.1 Hz. Overall, sway was increased when listening to music with 0.1 Hz volume shifting. Error bars indicate the standard deviation.

conditions and not because of a shifting in loudness. Researchers and clinicians using optic flow to examine sway may want to consider other forms of cueing such as changing the location of the sound source or leveraging the rhythm of the music to explore the interaction between visual and audio stimuli.

## Acknowledgments

The authors would like to thank Chauncey Frend and Jeff Rogers from the Indiana University UITS Advanced Visualization Lab for their efforts to support the hardware and software used in this study.

## Author Contributions

**Conceptualization:** Shaquitta Dent, Kelley Burger, Benjamin D. Smith, Jefferson W. Streepey.

**Data curation:** Shaquitta Dent, Jefferson W. Streepey.

**Formal analysis:** Shaquitta Dent, Jefferson W. Streepey.

**Investigation:** Shaquitta Dent, Kelley Burger, Skyler Stevens, Jefferson W. Streepey.

**Methodology:** Shaquitta Dent, Kelley Burger, Benjamin D. Smith, Jefferson W. Streepey.

**Project administration:** Shaquitta Dent, Jefferson W. Streepey.

**Resources:** Jefferson W. Streepey.

**Software:** Benjamin D. Smith.

**Supervision:** Jefferson W. Streepey.

**Validation:** Shaquitta Dent, Jefferson W. Streepey.

**Visualization:** Jefferson W. Streepey.

**Writing – original draft:** Shaquitta Dent, Kelley Burger.

**Writing – review & editing:** Shaquitta Dent, Skyler Stevens, Jefferson W. Streepey.

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
