## [Decision Letter · Decision Letter 0]

28 Apr 2021

PONE-D-21-02854

The effect of music on body sway when standing in a moving virtual environment

PLOS ONE

Dear Dr. Streepey,

Thank you for submitting your manuscript to PLOS ONE. After careful consideration, we feel that it has merit but does not fully meet PLOS ONE’s publication criteria as it currently stands. Therefore, we invite you to submit a revised version of the manuscript that addresses the points raised during the review process.

Two experts in the field have carefully reviewed the manuscript entitled, "The effect of music on body sway when standing in a moving virtual environment". Their comments are appended below.

The reviewer #1 gave detailed concerns on this manuscript standing from the professional viewpoint. I am sure his/her critiques are quite valuable to strengthen the manuscript.

The reviewer #2 raised several important concerns which should be considered before publication.

https://www.editorialmanager.com/pone/ and select the 'Submissions Needing Revision' folder to locate your manuscript file.

We look forward to receiving your revised manuscript.

Kind regards,

Manabu Sakakibara, Ph.D.

Academic Editor

PLOS ONE

Journal Requirements:

2. Please change "female” or "male" to "woman” or "man" as appropriate, when used as a noun (see for instance https://apastyle.apa.org/style-grammar-guidelines/bias-free-language/gender).

Reviewers' comments:

Reviewer's Responses to Questions

**Comments to the Author**

1. Is the manuscript technically sound, and do the data support the conclusions?

Reviewer #1: Partly

Reviewer #2: Yes

2. Has the statistical analysis been performed appropriately and rigorously? 

Reviewer #1: Yes

Reviewer #2: Yes

3. Have the authors made all data underlying the findings in their manuscript fully available?

Reviewer #1: Yes

Reviewer #2: Yes

4. Is the manuscript presented in an intelligible fashion and written in standard English?

Reviewer #1: Yes

Reviewer #2: Yes

5. Review Comments to the Author

Reviewer #1: Standing participants wore a head-mounted display. They viewed a stationary scene, or one that oscillated along the line of sight. The headphones were silent, or played a musical selection, which had constant volume, or volume that increased and decreased in an oscillatory fashion. The authors evaluated measures of the spatial magnitude of postural sway relative to the Earth.

Could subjects simply have been moving their body with the beat of the music? Such an effect would be uninteresting, given the propensity to tap our feet in time with music. It would be helpful to analyze the postural data for frequency peaks that might be related to the rhythm of the music.

For a broader consideration of the control of the body relative to the acoustic environment, it would be helpful to cite studies on music/rhythm influence on gait (e.g., Dalla Bella et al., 2018; Hunt et al., 2014), as well as responses of blind people to acoustic oscillation (Stoffregen et al., 2010).

The authors elected to oscillate the loudness of music. The stated motivation for this choice is empirical: It has “worked” in previous studies. No explanation is offered as to how this particular variation could “enhance the perception of visual motion”. Stronger motivation might be found for the use of sound (music, or any other) that varied in 3-d localization; specifically, moving sound fields simulating fore-aft translation. Stoffregen et al. (2009; see also Stoffregen et al., 2010) found that postural control was influenced by sound fields corresponding to 3-d displacement of the subject relative to the Earth.

The focus on the magnitude of sway appears to be sub-optimal. Given the use of oscillatory stimulus motion, and the inherently oscillatory nature of sway, it would seem more important to evaluate the extent to which sway was coupled with the motion stimulus.

The authors have information about sway, and they have information about the oscillation of their optic and acoustic stimuli. Thus, they have the opportunity to measure coupling, for example, using AMI (Stoffregen et al., 2009), or cross-coherence (e.g., Varlet et al., 2015). It would be interesting to compare sway-optics coupling vs. sway-acoustics coupling, and to compare coupling across the different frequency conditions.

Adding a measure of coupling could make it easier for the authors to reduce or eliminate some of their current dependent variables. Because all are derived from data on the spatial magnitude of sway, excursion, RMS, peak velocity, and mean velocity are correlated. That is, they are not orthogonal to each other. For this reason, the effects reported for these dependent variables cannot be interpreted as being independent of one another. Of these, excursion and peak velocity are the least informative (because they provide information only about transient extremes in the data); I suggest that these be dropped.

The data figures are difficult to interpret because the authors have not defined the error bars that are shown. Do the error bars illustrate the standard deviation of the mean? The standard error? Something else?

In the Discussion, the authors state, “the sensation of body motion associated with the translating VR scene led to increased body sway for all AP measures”. This statement is very puzzling, as the authors do not report any data on participants’ subjective experience. Without such data, it is not possible to infer that any aspect of subjective experience “led to” increased sway. The authors may care to speculate that their postural effects co-occurred with some subjective experience, but it is important to bear in mind the many studies showing that postural sway is influenced by imposed motion even when participants have no subjective experience of motion (e.g., Stoffregen, 1985). The experience of self-motion is not a prerequisite for the influence of imposed motion on the control of stance.

Dalla Bella, Dotov, Bardy, & Cochen de Cock (2018). Individualization of music-based rhythmic auditory cueing in Parkinson’s disease. Annals of the New York Academy of Sciences, doi: 10.1111/nyas.13859

Hunt, McGrath, & Stergiou (2014). The influence of auditory-motor coupling on fractal dynamics in human gait. Scientific Reports, 4, article #5879.

Stoffregen, T. A. (1985). Flow structure versus retinal location in the optical control of stance. Journal of Experimental Psychology: Human Perception and Performance, 11, 554-565.

Stoffregen, T. A., Ito, K., Hove, P., Yank, J. R., & Bardy, B. G. (2010). The postural responses of adults who are blind to a moving environment. Journal of Visual Impairment and Blindness, 104, 73-83.

Stoffregen, T. A., Villard, S., Kim, C., Ito, K., & Bardy, B. G. (2009). Coupling of head and body movement with motion of the audible environment. Journal of Experimental Psychology: Human Perception & Performance, 35, 1221-1231.

Varlet, M., Bardy, B. G., Chen, F.-C., Alcantara, C., & Stoffregen, T. A. (2015). Coupling of postural activity with motion of a ship at sea. Experimental Brain Research, 233, 1607-1616.

Reviewer #2: Overall, this study seems to be a valuable contribution to perception literature. The reviewers main take-away is that the authors intent was to explore the influence of environmental motion as well as auditory stimulus to better understand their role on postural sway and ultimately perception/physical adjustments to our surroundings.

Lines in manuscript:

49-51) “When the supplied …. sway.”- This could be phrased a bit clearer.

58) The term translation is used throughout the paper to describe the VR motion, it would be helpful to clearly define this term early on for readers that may be less familiar with its use here.

62) The term “college-aged” is very vague. “College students” would be more appropriate.

SET-UP

90-91) It should be noted if participants in the eyes closed condition saw the same VR environment as other conditions before being asked to close their eyes.

93) The authors should give their reasoning for manipulating the environment along the AP plane alone.

96-97) It would be useful to note if volume selection made by the subjects influenced the outcome of noise/music.

98) Was there any type of white noise played in the no music condition?

99-102) The reviewers understanding of this is that as the scene switched from anterior to posterior movement the volume also adjusted. If this is the case it would be worthwhile to describe it in clearer detail.

128/130) The reviewer is unsure of the purpose of question marks following the word with in both these lines.

STATISTICS-

In general, the statistics seem sound. However, an additional analysis that the authors should consider is to translate the COP data to a time series and calculate a type of complexity measure such as Effort to Compress or Multifractal Spectrum Width. It would give the authors a more detailed look at overall movement and motion complexity rather than just peak excursions, peak velocities, and average velocities along the AP and ML planes.

Here are some references:

Masoner, H., Hajnal, A., Clark, J. D., Dowell, C., Surber, T., Funkhouser, A., ... & Wagman, J. B. (2020). Complexity of postural sway affects affordance perception of reachability in virtual reality. Quarterly Journal of Experimental Psychology, 1747021820943757.

Nagaraj, N., & Balasubramanian, K. (2017). Dynamical complexity of short and noisy time series. The European Physical Journal Special Topics, 226(10), 2191-2204.

Nagaraj, N., & Balasubramanian, K. (2017). Three perspectives on complexity: entropy, compression, subsymmetry. The European Physical Journal Special Topics, 226(15), 3251-3272.

With a sampling rate of 1000 hz there should be enough data points to perform a multifractal analysis:

Hajnal, A., Clark, J. D., Doyon, J. K., & Kelty-Stephen, D. G. (2018). Fractality of body movements predicts perception of affordances: Evidence from stand-on-ability judgments about slopes. Journal of Experimental Psychology: Human Perception and Performance, 44(6), 836.

DISCUSSION-

The authors should address the fact that the VR translation was along the AP plane and that this manipulation overlaps with all significant effects. Since there were no significant effects reported along the ML plane.

6. PLOS authors have the option to publish the peer review history of their article (what does this mean?). If published, this will include your full peer review and any attached files.

Reviewer #1: No

Reviewer #2: No

---

## [Author Response · Author response to Decision Letter 0]

29 Aug 2021

We have also uploaded this response as a document.

Dear editor and reviewers,

Below are our responses to the thoughtful suggestions made by each of you. We have attempted to incorporate your feedback as best as we could and believe that our manuscript is greatly enhanced by your review. We are looking forward to any further feedback or suggestions that you may have.

The style of the manuscript has been changed to meet the requirements. References are cited in order at first mention. 

2. Please change "female” or "male" to "woman” or "man" as appropriate, when used as a noun (see for instance https://apastyle.apa.org/style-grammar-guidelines/bias-free-language/gender).

This has been changed in the manuscript (Line 64).

Reviewer #1: Standing participants wore a head-mounted display. They viewed a stationary scene, or one that oscillated along the line of sight. The headphones were silent, or played a musical selection, which had constant volume, or volume that increased and decreased in an oscillatory fashion. The authors evaluated measures of the spatial magnitude of postural sway relative to the Earth.

Could subjects simply have been moving their body with the beat of the music? Such an effect would be uninteresting, given the propensity to tap our feet in time with music. It would be helpful to analyze the postural data for frequency peaks that might be related to the rhythm of the music.

The authors agree that this it is entirely possible that the subjects were swaying to the music and that the sway frequency could be related to the music. We have introduced this possibility into the discussion (lines 188-193). However, at this point in time there is no one on the research team with a background in music who is available to help with this analysis.

For a broader consideration of the control of the body relative to the acoustic environment, it would be helpful to cite studies on music/rhythm influence on gait (e.g., Dalla Bella et al., 2018; Hunt et al., 2014), as well as responses of blind people to acoustic oscillation (Stoffregen et al., 2010).

The authors noted this suggestion and reviewed the papers suggested by the reviewer. Although they were not all included in this revision, they did point us to other literature that was cited in the manuscript and overall helped to shape the discussion section. 

The authors elected to oscillate the loudness of music. The stated motivation for this choice is empirical: It has “worked” in previous studies. No explanation is offered as to how this particular variation could “enhance the perception of visual motion”. Stronger motivation might be found for the use of sound (music, or any other) that varied in 3-d localization; specifically, moving sound fields simulating fore-aft translation. Stoffregen et al. (2009; see also Stoffregen et al., 2010) found that postural control was influenced by sound fields corresponding to 3-d displacement of the subject relative to the Earth.

Moving the sounds fields is a great idea. The authors just weren’t thinking in that direction. We were concurrently doing an experiment were we manipulated the pitch, loudness, and frequency of the movement and were finding little difference among the 3 (unfortunately the students doing this study did not finish). We settled on using loudness because of the 3 we felt that that condition would be the easiest to replicate should anyone ever want to reproduce the methods of this study.

References the works cited by the reviewer have been added to the discussion of the present study’s limitations. (Lines 180-185)

The focus on the magnitude of sway appears to be sub-optimal. Given the use of oscillatory stimulus motion, and the inherently oscillatory nature of sway, it would seem more important to evaluate the extent to which sway was coupled with the motion stimulus.

The authors have information about sway, and they have information about the oscillation of their optic and acoustic stimuli. Thus, they have the opportunity to measure coupling, for example, using AMI (Stoffregen et al., 2009), or cross-coherence (e.g., Varlet et al., 2015). It would be interesting to compare sway-optics coupling vs. sway-acoustics coupling, and to compare coupling across the different frequency conditions.

The authors appreciated this idea and have examined the coherence of the data. The Methods, Results, and Discussion have all be updated to report on this new analysis.

Of these, excursion and peak velocity are the least informative (because they provide information only about transient extremes in the data); I suggest that these be dropped.

These have been dropped as suggested

The data figures are difficult to interpret because the authors have not defined the error bars that are shown. Do the error bars illustrate the standard deviation of the mean? The standard error? Something else?

It should have been noted that these were standard deviations. The figure captions been updated to clearly state that the means and standard deviations are being displayed. (Lines 140-141 and 156)

In the Discussion, the authors state, “the sensation of body motion associated with the translating VR scene led to increased body sway for all AP measures”. This statement is very puzzling, as the authors do not report any data on participants’ subjective experience. Without such data, it is not possible to infer that any aspect of subjective experience “led to” increased sway. The authors may care to speculate that their postural effects co-occurred with some subjective experience, but it is important to bear in mind the many studies showing that postural sway is influenced by imposed motion even when participants have no subjective experience of motion (e.g., Stoffregen, 1985). The experience of self-motion is not a prerequisite for the influence of imposed motion on the control of stance.

Again, the authors appreciate this comment and believe that the reviewer was right to point it out. This paragraph has been re-written to remove this confusing language. The sentence now reads:

As expected, the translating VR scene led to increased body sway in the AP direction that was strongly correlated with the visual motion stimulus. Lines 160-161

Dalla Bella, Dotov, Bardy, & Cochen de Cock (2018). Individualization of music-based rhythmic auditory cueing in Parkinson’s disease. Annals of the New York Academy of Sciences, doi: 10.1111/nyas.13859

Hunt, McGrath, & Stergiou (2014). The influence of auditory-motor coupling on fractal dynamics in human gait. Scientific Reports, 4, article #5879.

Stoffregen, T. A. (1985). Flow structure versus retinal location in the optical control of stance. Journal of Experimental Psychology: Human Perception and Performance, 11, 554-565.

Stoffregen, T. A., Ito, K., Hove, P., Yank, J. R., & Bardy, B. G. (2010). The postural responses of adults who are blind to a moving environment. Journal of Visual Impairment and Blindness, 104, 73-83.

Stoffregen, T. A., Villard, S., Kim, C., Ito, K., & Bardy, B. G. (2009). Coupling of head and body movement with motion of the audible environment. Journal of Experimental Psychology: Human Perception & Performance, 35, 1221-1231.

Varlet, M., Bardy, B. G., Chen, F.-C., Alcantara, C., & Stoffregen, T. A. (2015). Coupling of postural activity with motion of a ship at sea. Experimental Brain Research, 233, 1607-1616.

Reviewer #2: Overall, this study seems to be a valuable contribution to perception literature. The reviewers main take-away is that the authors intent was to explore the influence of environmental motion as well as auditory stimulus to better understand their role on postural sway and ultimately perception/physical adjustments to our surroundings.

Lines in manuscript:

49-51) “When the supplied …. sway.”- This could be phrased a bit clearer.

This has been re-written as follows

When the provided auditory cue was music as opposed to white noise or tones, Palm et al. [8] showed that body sway was not differently affected. (Lines 50-51)

58) The term translation is used throughout the paper to describe the VR motion, it would be helpful to clearly define this term early on for readers that may be less familiar with its use here.

Translation has been better defined in this sentence

For this study we used sound to enhance the perception of visual motion as a VR environment translated in the subject’s sagittal plane so that the environment appeared to move forward and backward about the subject. (Lines 56-58)

62) The term “college-aged” is very vague. “College students” would be more appropriate.

Changed. (Line 64)

SET-UP

90-91) It should be noted if participants in the eyes closed condition saw the same VR environment as other conditions before being asked to close their eyes.

They did. The head mounted display was not removed for these conditions and the scene was always shown. This would be no different from opening and closing your eyes in a regular room. The manuscript has been updated to make this clear. 

During this time the subjects only viewed the virtual environment including when they were between trials (Lines 93-94) 

93) The authors should give their reasoning for manipulating the environment along the AP plane alone.

This reasoning has been included in the methods.

AP translation of the scene was chosen because it has been well documented that such visual motion can invoke a postural response [13,14]. (Lines 98-99)

Other directions, such a medial lateral, yaw, and roll, were considered to be collected along with the AP translation; however in the interest of time and the comfort of the subjects the experiment was reduced to just AP motion. 

96-97) It would be useful to note if volume selection made by the subjects influenced the outcome of noise/music.

Agreed. This is a good point. However we do not have a record the volumes used. 

98) Was there any type of white noise played in the no music condition?

No noise was provided. The subject were simply in a quiet room. The manuscript has been revised to indicate this

There was also a fourth audio condition where no music was played and no other sounds were provided. (Lines 103-104)

99-102) The reviewers understanding of this is that as the scene switched from anterior to posterior movement the volume also adjusted. If this is the case it would be worthwhile to describe it in clearer detail.

The methods have been updated to make sure that it is understood that the volume scaled up as the scene translated in a way consistent with forward movement and scaled down as the scene moved back. The manuscript has been revised to make this clear

…the signals started simultaneously so that the reversal points of the 0.1 Hz music volume shifting matched the reversal points of the scene translation and that the volume scaled up as the subjects viewed forward motion and the volume scaled down as backwards motion was viewed. (Lines 106-109)

128/130) The reviewer is unsure of the purpose of question marks following the word with in both these lines.

Questions marks have been removed.

STATISTICS-

In general, the statistics seem sound. However, an additional analysis that the authors should consider is to translate the COP data to a time series and calculate a type of complexity measure such as Effort to Compress or Multifractal Spectrum Width. It would give the authors a more detailed look at overall movement and motion complexity rather than just peak excursions, peak velocities, and average velocities along the AP and ML planes.

At the suggestion of reviewer 1, a cross-coherence analysis of the data has been performed. We hope that this also satisfies the request of reviewer #2 to provide a more detailed look at the movement’s complexity. 

Here are some references:

Masoner, H., Hajnal, A., Clark, J. D., Dowell, C., Surber, T., Funkhouser, A., ... & Wagman, J. B. (2020). Complexity of postural sway affects affordance perception of reachability in virtual reality. Quarterly Journal of Experimental Psychology, 1747021820943757.

Nagaraj, N., & Balasubramanian, K. (2017). Dynamical complexity of short and noisy time series. The European Physical Journal Special Topics, 226(10), 2191-2204.

Nagaraj, N., & Balasubramanian, K. (2017). Three perspectives on complexity: entropy, compression, subsymmetry. The European Physical Journal Special Topics, 226(15), 3251-3272.

With a sampling rate of 1000 hz there should be enough data points to perform a multifractal analysis:

Hajnal, A., Clark, J. D., Doyon, J. K., & Kelty-Stephen, D. G. (2018). Fractality of body movements predicts perception of affordances: Evidence from stand-on-ability judgments about slopes. Journal of Experimental Psychology: Human Perception and Performance, 44(6), 836.

DISCUSSION-

The authors should address the fact that the VR translation was along the AP plane and that this manipulation overlaps with all significant effects. Since there were no significant effects reported along the ML plane.

The discussion has been updated to make this clear.

As expected, the translating VR scene led to increased body sway in the AP direction that was strongly correlated with the visual motion stimulus. This finding is in line with previous studies examining the effect of AP optic flow on body sway [13,14] and is consistent with the dorsiflexor and plantarflexor motion of the ankle observed during stance with the feet placed in a neutral position [18]. (Lines 160-164).

---

## [Decision Letter · Decision Letter 1]

6 Sep 2021

**Comments to the Author**

1. If the authors have adequately addressed your comments raised in a previous round of review and you feel that this manuscript is now acceptable for publication, you may indicate that here to bypass the “Comments to the Author” section, enter your conflict of interest statement in the “Confidential to Editor” section, and submit your "Accept" recommendation.

Reviewer #1: All comments have been addressed

Reviewer #2: All comments have been addressed

2. Is the manuscript technically sound, and do the data support the conclusions?

Reviewer #1: Yes

Reviewer #2: Yes

3. Has the statistical analysis been performed appropriately and rigorously? 

Reviewer #1: Yes

Reviewer #2: Yes

4. Have the authors made all data underlying the findings in their manuscript fully available?

Reviewer #1: Yes

Reviewer #2: Yes

5. Is the manuscript presented in an intelligible fashion and written in standard English?

Reviewer #1: Yes

Reviewer #2: Yes

6. Review Comments to the Author

Reviewer #1: Well done well done well done well done the revision is adequate I am trying to achieve the required minimum character limit. Gosh, a great many characters appear to be reaquired for me to say that they did a good job in revising. This requirement is both pointless and punitive, as I am nearing the point at which I give up and delete my review entirely. Am I getting closer in communicating that the revision is adequare? I guess not; the Red Letters of Death remain red.

Reviewer #2: The authors seem to have addressed all reviewers comments.

There is a small grammatical error on line 121 "Coherence estimates close 1.0" should read: Coherence estimates close to 1.0.

Aside from this minor change the manuscript seems ready for publication.

7. PLOS authors have the option to publish the peer review history of their article (what does this mean?). If published, this will include your full peer review and any attached files.

Reviewer #1: **Yes: **Thomas A. Stoffregen

Reviewer #2: No

---

## [Author Response · Author response to Decision Letter 1]

14 Sep 2021

Below is our response to the issue brought forth by Reviewer #2. We appreciate all of the time that both reviewers took to provide thoughtful feedback.

Reviewer #2: There is a small grammatical error on line 121 "Coherence estimates close 1.0" should read: Coherence estimates close to 1.0.

Line 121 has been changed to read “Coherence estimates close to 1.0...”

---

## [Editor Report · Decision Letter 2]

16 Sep 2021

The effect of music on body sway when standing in a moving virtual environment

PONE-D-21-02854R2

Dear Dr. Streepey,

We’re pleased to inform you that your manuscript has been judged scientifically suitable for publication and will be formally accepted for publication once it meets all outstanding technical requirements.

Kind regards,

Manabu Sakakibara, Ph.D.

Academic Editor

PLOS ONE
---

## [Editor Report · Acceptance letter]

20 Sep 2021

PONE-D-21-02854R2 

The effect of music on body sway when standing in a moving virtual environment 

Dear Dr. Streepey:

I'm pleased to inform you that your manuscript has been deemed suitable for publication in PLOS ONE. Congratulations! Your manuscript is now with our production department. 

Kind regards, 

on behalf of

Dr. Manabu Sakakibara 

Academic Editor

PLOS ONE